**Data Availability Statement:** All relevant data are within the manuscript and its Supporting Information files.

**Funding:** This research was financed by Addis Ababa University. However, the funder had not any

# Medical laboratory waste generation rate, management practices and associated factors in Addis Ababa, Ethiopia

**Salem Endris, Zemenu Tamir, Abay Sisay**📷*

Department of Medical Laboratory Sciences, College of Health Science, Addis Ababa University, Addis Ababa, Ethiopia

* abusis27@gmail.com

## Abstract

### Background

Biomedical wastes (BMWs) generated from medical laboratories are hazardous and can endanger both humans and the environment. Highly infectious biomedical wastes are produced at an unacceptably high rate from health laboratories in developing countries with poor management systems, such as Ethiopia. The purpose of this study was to assess the rate of biomedical waste generation, management practices, and associated factors in public healthcare medical laboratories in Addis Ababa, Ethiopia.

### Materials and methods

From July 13 to September 25, 2020, a health institution-based cross-sectional study was conducted in 6 hospital laboratories and 20 health centres laboratories in Addis Ababa, Ethiopia. Data on socio-demographic characteristics, knowledge and practice of biomedical waste management and generation rate were collected d in health facilities using pre tested data collection tools. SPSS version 20 was used to manage the data. To identify independent predictors of the dependent variable, descriptive statistics, Pearson correlation, linear, and logistic regression analysis were used. The strength of the association was determined using an odds ratio with a 95% confidence interval.

### Results

In this study, the mean ± SD daily generation rate of biomedical wastes was 4.9 ± 3.13 kg/day per medical laboratory. Nineteen medical laboratories (74.3%) had proper biomedical waste management practice, which is significantly associated with professionals' knowledge of biomedical waste management policies and guidelines, the availability of separate financial sources for biomedical waste management, and the level of training of professionals.

### Conclusion

The study found that medical laboratories in Addis Ababa's public healthcare facilities generate a significant amount of biomedical waste. Nearly two-thirds of hospitals performed

involvement with the research methodology design, analysis and write up of the manuscript.

**Competing interests:** The authors have declared that no competing interests exist.

proper waste segregation, collection, storage, and treatment procedures for biomedical waste generated in their laboratories. However, there was a poor transportation and disposal method. As a result, paying special attention and implementing the current national guidelines for biomedical waste management is recommended.

## Introduction

Biomedical waste contains all wastes generated by health care facilities, research facilities, and clinical laboratories during the diagnosis, treatment, or immunization of humans, as well as wastes generated in related research activities that pose a greater risk to human health than any other wastes [1, 2]. Biomedical wastes are generated in large quantities by public health and diagnostic laboratories. Medical laboratories generate considerable amounts of hazardous wastes that require particular packaging, handling, and treatment methods as a result of the growing demand for medical services and technology [3, 4]. Laboratories generate a higher volume of chemical wastes, clinical glasses, culture plates, stock cultures, highly infectious wastes in huge amounts, and certain radioactive wastes [1, 5].

Inappropriate management of these hazardous biomedical wastes results in exposing staffs in health care facilities, patients, waste collectors and the public in general to infections pathogens [3]. Infectious healthcare wastes can transmit more than 30 dangerous blood borne pathogens, but the pathogens that are found to be significant are hepatitis B, hepatitis C, and Human immune deficiency virus (HIV). Needle stick injuries are expected to result in 21 million hepatitis B, 2 million hepatitis C, and 260,000 HIV infections each year around the world [6, 7]. Proper management of infectious biomedical wastes should not be optional rather mandatory [8]. Hence, appropriate healthcare waste management with its vital steps; control of generation, segregation, collection, storage, transport, treatment, and disposal in a manner that follows best principles of public health, economics, engineering, conservation, aesthetics, and other environmental considerations is very crucial [2, 6].

Globally, 2 million healthcare workers are exposed of infection [9]. The potential risks posed by BMWs' unsafe management and disposal have long been recognized in the Western Pacific Region, and efforts have been made to raise awareness of the issue [10]. Nonetheless, knowledge and practice of treating infectious wastes by health care professionals working in developing countries differ from that of developed countries. According to a systematic review of studies conducted on the African continent, 47% of the studies stated that waste segregation is underwhelming due to lack of segregation utilities, lack of awareness, and enforcing laws/ regulations [11].

According to an Ethiopian systematic review, the proportion of hazardous wastes generated in Ethiopian healthcare facilities were unacceptably high, ranging from 21% to 70%, due to very limited waste segregation practices in studied facilities and reviewed studies [4]. It has been demonstrated that the absence of policies or regulations regarding biomedical waste management, lack of awareness and inadequate training of professionals, lack of human resources and financial scarcity in many health facilities are all associated with poor waste management and lack of protective measures [6, 10, 12, 13]. However, the rate of waste generation and waste management practices in Ethiopian medical laboratories, which are important sources of infectious waste, are not well documented. As a result, this study was conducted to assess the generation rate and management system of biomedical wastes of medical laboratories and its associated factors in Addis Ababa, Ethiopia, which helps in identifying factors

affecting biomedical waste management (BMWM), initiating stakeholders to amend or develop more extensive policies and strategies, and alerting laboratory managers to pay attention to it.

## Materials and methods

### Study design, setting, and participants

A cross-sectional study was conducted in selected public healthcare facility laboratories (Health centers and Hospitals) in Addis Ababa, Ethiopia from July to September 2020. Addis Ababa is Ethiopia's largest and capital city, with a population of 6.5 million and a land area of 527 square kilometres divided into ten sub-cities [14]. Addis Ababa city administration oversees 99 health centers and six public health hospitals. Each hospital and health center serves between 1 and 1.5 million and 40,000 people [15, 16].

   The study included a total of 26 public health facilities which encompass six public hospitals and twenty health centres, selected by simple random sampling method. The number of health centres to be included from each sub-city were determined by allocating them in proportion to their population. To select study participants (medical laboratory professionals and waste handlers) from each public health facility, convenience sampling method was used. During the data collection period, 25–26 laboratory professionals and 4 waste handlers from each public hospital, as well as 7 laboratory professionals and 2 waste handlers from each health centre were enrolled. Moreover, medical laboratory managers and quality officers were selected purposively and included in the study.

### Data collection procedure

Primary data was collected and recorded using a data collection tool specifically designed for this study. For seven consecutive days, solid biomedical wastes generated by each laboratory were measured daily using a daily pre-calibrated weighing scale (kg/day). Empty plastic bags of a standard color were provided, and each was labelled with the sample number, date of collection, place of generation, and type of waste.

   A self-administered questionnaire (**S1 File**) developed by reviewing literature, articles, international and national biomedical waste management guidelines, and assessment tools [1–3, 9, 11, 13, 17–19] was used to assess biomedical waste management practices and related factors. The questionnaire consists of three sections: socio-demographic characteristics, laboratory professional knowledge of BMWM, healthcare facilities BMWM practice, and associated factors. To assess the practice of biomedical waste management, a separate questionnaire for waste handlers was developed in English and then translated into Amharic. In order to assess the actual use of the BMWM system, an observational checklist and an interview were used. Before beginning the actual data collection, the data collection tool was pretested in 5% of the sample. For three months, two trained data collectors collected data under the supervision of the principal investigators, and both Amharic and English were used as data collection mediums.

### Operational definition of terms

   **Biomedical waste.**   A waste generated from medical laboratories during an investigation of body fluids like blood, urine, stool, sputum, and other body fluids which could be hazardous or non–hazardous.

   **Hazardous biomedical waste.**   Is defined as waste produced by medical laboratories that pose a significant risk to public health and the environment.

**Infectious waste.**   All waste and tools that could be contaminated with blood or other human fluids, such as contaminated gloves, swabs, cotton, sputum cups, and slides.

**Sharp waste.**   Is such as needles, lancets, blood collection and infusion sets, and shattered glassware, is described as infectious trash that can pierce the skin.

**Non-hazardous biomedical waste.**   Waste that has not been contaminated with blood or bodily fluids.

**Biomedical waste management system.**   Is a system that controls the generation, segregation, collection, storage, transport, treatment, and disposal of biomedical wastes generated from medical laboratories in health care facilities [1–3, 10, 17, 20].

**Proper practice.**   Is a public health facility that performs more than three components of biomedical waste management system accordingly to national policy and guideline of health care waste management in Ethiopia.

**Poor practice.**   Means a public health facility that practices less than three components of the biomedical waste management system.

## Data analysis and interpretation

For analysis, data were entered into a statistical package for social science (SPSS) version 20. Tables were used to present data that was summarized using descriptive statistics. To identify independent predictors of the dependent variable, Pearson correlation, linear, bi variable, and multivariable logistic regression analysis were used. The strength of the association was determined using an odds ratio with a 95% confidence interval.

## Ethical consideration

The research ethics review committee of the department of Medical Laboratory Science, College of Health Sciences, Addis Ababa University, granted ethical clearance (Ref. #-DRERC/552/20/MLS) and a letter of request was sent to the Addis Ababa Health Bureau. Six public hospitals and ten sub-cities received official letters from Addis Ababa's public health research and emergency management directorate. The respective health offices of each sub-city wrote permission letters for the study to be conducted in selected health centres. After being briefed on the purpose and significance of the study, each study participant provided informed consent. To maintain the anonymity of the study participants, personal identifiers were removed and only codes were used throughout the study.

## Results

### Socio-demographic characteristics

A total of 362 participants (298 medical laboratory professionals and 64 waste handlers) from 26 healthcare facilities were included in this study. There were a total of 194 (53.6%) females, with 130 (43.6%) laboratory professionals and 64 (100%) waste handlers. The respondents' mean ± SD age was 30.4 ± 6.63 years. In terms of educational attainment, 212 (58.6%) and 84 (23.2%) study participants held Bachelor's degree and Diploma, respectively. Among all, 152 (42%) and 130 (35.9%) had work experience ranging from 5 to 10 years and 1 to 5 years, respectively [Table 1].

**Knowledge of study participants about the biomedical waste management.**   The majority of laboratory professionals (63%) and waste handlers (81.3%) who took part in this study were aware of Ethiopia's biomedical waste management policies and guidelines. In terms of storage time, 198(54.7%), 74(20.4%), 20(5.5%), and 6(1.7%) laboratory professional were aware that the maximum time for storing biomedical waste in the facility is 12–24 hours, less

**Table 1. Socio-demographic characteristics of study participants in Addis Ababa, Ethiopia (n = 362).**

| Socio-demographic character | Variable | | | | | Total Percent (n = 362) |
|---|---|---|---|---|---|---|
| | | Laboratory professionals (n = 298) | | Waste handlers (n = 64) | | |
| | | Frequency | Percent | Frequency | Percent | |
| Sex | Male | 168 | 56.4% | | | 46.4% |
| | Female | 130 | 43.6% | 64 | 100% | 53.6% |
| Age | <21 years | 8 | 2.7% | 18 | 28.1% | 7.1% |
| | 21–30 years | 186 | 62.4% | 24 | 37.5% | 58% |
| | 31–40 years | 97 | 32.5% | 18 | 28.1% | 31.7% |
| | > 40 years | 12 | 3.9% | 4 | 6.3% | 4.4% |
| Level of Education | 1-8th grade | | | 16 | 25% | 4.4% |
| | 9-12th grade | | | 40 | 62.5% | 11% |
| | Diploma | 82 | 27.5% | 2 | 3.1% | 23.2% |
| | BSC degree | 208 | 69.8% | 4 | 6.3% | 58.6% |
| | MSC | 8 | 2.7% | | | 2.2% |
| | other | | | 2 | 3.1% | .5% |
| Work experience | <1 year | 20 | 6.7% | 10 | 15.6% | 8.3% |
| | 1–5 years | 98 | 32.9% | 32 | 50% | 35.9% |
| | 5–10 years | 134 | 45% | 18 | 28.1% | 42% |
| | 10–15 years | 34 | 11.4% | 2 | 3.1% | 9.9% |
| | 15–20 years | 10 | 3.4% | 2 | 3.1% | 2.8% |
| | >20 years | 2 | 0.7% | | | 1.1% |

than 12 hours, 24–48 hours, and greater than 48 hours, respectively. Only 40 (11%) study participants stated that incineration is the most common mode of treatment of biomedical wastes before final disposal, while the remaining 128 (35.4%) stated that bleaching is the most common mode of treatment. More than three fourth (78.5%) of study participants were aware that the safety box must be discarded when 3/4 was full. Half of the laboratory professionals (50.3%) and half of the waste handlers (90.6%) had received biomedical waste management training [Table 2].

**Table 2. Frequency of laboratory professionals of response for knowledge item questions for a study of biomedical waste management, Addis Ababa, Ethiopia (n = 298).**

| Variable | Frequency Yes | Percent |
|---|---|---|
| Knowledge about biomedical waste management policy and guidelines | 228 | 63% |
| Knowledge about the common types of biomedical waste | 92 | 30.9% |
| Knowledge about the basic components of biomedical waste management | 92 | 30.9% |
| Knowledge about the common infectious microorganisms transmitted during biomedical waste management | 162 | 54.4% |
| Knowledge about the common health hazards associated with poor biomedical waste management | 148 | 49.7% |
| Knowledge about the maximum time biomedical wastes storage | 198 | 66.4% |
| Knowledge about the color code of waste bins that suit a type of biomedical waste | 284 | 95.3% |
| Knowledge about the common modes of treatment of biomedical waste before final disposal | 40 | 13.4% |
| Knowledge about the appropriate method for final disposal of biomedical wastes | 40 | 13.4% |
| Knowledge about safety box disposal time | 284 | 95.3% |
| Received training about the biomedical waste management | 150 | 50.3% |

**Biomedical waste management practice.**   Two hundred ninety-six (99.3%) of medical laboratory professionals responded that their laboratory has separate containers for the collection of hazardous and non-hazardous biomedical wastes. Among these, 286 (96%) confirmed that they used color coding based segregation, while the remaining 10 (3.4%) do not use color coding as segregation, and 264 (88.6%) waste containers have a biohazard symbol. Four healthcare facilities (16%) do not practice proper segregation. Approximately 56.3%, 28.1%, and 15.6% of waste handlers confirmed that they collect biomedical wastes from the laboratory every 12 hours, 8 hours, and 24 hours, respectively. Concerning sharp waste management, 276 (92.6%) medical laboratory professionals stated that safety boxes are available in arm reach locations for sharp waste collection and are collected when they are 3/4 filled, according to 94.6% of participants. Waste holding bags and containers are durable enough, according to 256 (85.9%) medical laboratory professionals and 63 (96.9%) waste handlers. Personal protective equipment was used during waste collection in all health care facilities. However, 79.2% of medical laboratory professionals and 90.6% of waste handlers reported that their facility provides personal protective equipment. On the other hand, 18.2% of laboratory professionals have reported as they encountered physical injuries like needle stick injury and sharp injury.

Relating the precaution and method of transportation of biomedical wastes, 48.3%, 23.5%, 18.1%, and 8.1% of medical laboratory professionals indicated that the methods used for transportation of biomedical wastes are holding the waste-collecting bags with bare hands, use of wheelbarrows, use of the trolley, and by the waste container itself, respectively. Twenty-six (40.6%) waste handlers also confirmed that their facility does not have a separate biomedical waste transportation tool. Infectious waste collection bins had become covered, according to 63% of study participants, since the coronavirus pandemic.

According to 93.8% of waste handlers, the majority of healthcare facilities have storage areas isolated from medical equipment and café storerooms. Biomedical waste is stored for less than 12 hours, according to 78.1% of waste handlers. Approximately 70.2% of medical laboratory professionals confirmed that infectious wastes are treated before disposal using methods such as incineration, autoclaving, and chemical disinfection. According to 69.6% of medical laboratory professionals, liquid wastes are also decontaminated before being dumped into running water, whereas 11.6% said it is simply dumped into running water without being decontaminated.

In terms of waste disposal, 38.1%, 18.8%, and 12.2% of medical laboratory professionals indicated that open burning pits, landfills, and direct to the municipal waste system are the most common types of disposal methods in their facility, respectively; whereas 59.4%, 9.4%, and 13% of waste handlers responded that treated biomedical wastes are disposed of in open pits, landfills, and municipal waste systems, respectively.

In this study, 67.4% of study participants confirmed that the manager of their facility's medical laboratory is concerned about biomedical waste management as part of their routine work. According to the study, only 18.2% of study participants confirmed that their institution has a separate financial source for biomedical waste management, and 49.7% believed that their institution legitimately follows Ethiopia's current biomedical waste management guidelines. The study also found that 69.6% of medical laboratory professionals were vaccinated for both hepatitis B virus and tetanus, and 6.1% were only vaccinated for hepatitis B virus, respectively, while 6.6% were not vaccinated for both.

**Observational result.**   The researchers discovered that twenty (76.9%) of the facilities have the appropriate equipment for handling biomedical wastes, such as personal protective equipment, safety boxes, and waste containers. Eleven (42.3%), eight (30.7%), and six (23%), respectively, of these health care facilities, had both color-coded and labelled bins, only color-coded bins, and only labelled waste containers in their laboratory. Despite this, four facilities (15.3%)

do not follow proper segregation procedures. In 22 (84.6%) laboratories wastes related to the COVID-19 pandemic such as face masks were segregated in a waste collection bin with a cover. While the remaining 3 (11.5%) did not segregate in a waste collection bin with cover. In 17 (65.3%), 6 (23%), and 2 (7%) laboratories, wastes were collected twice a day, once a day, and three times a day, respectively. Personal protective equipment was used during waste collection in all health care settings.

In terms of transportation, 7 (26.7%) and 6 (23%) facilities used transportation trolleys and wheelbarrows to transport biomedical wastes respectively. In the other hand, 12 (46.1%) health facilities lacked transportation materials. Fourteen (56%) facilities had a waste storage area, whereas 11(42.3%) did not. Eleven (42.3%) of these are found far enough on the premises. However, 12 (85.7%) are open storage rooms and 2 (14.3%) are secured storage areas.

The majority of facilities used incineration as a means of biomedical waste treatment, with 23 (88.4%) incinerators being low temperature (made of brick and clay) and only 1 (3.8%) being high temperature (made of electrical system). Twenty two (84.6%) of the incinerators had adequate air inlet and outlet, whereas the remaining 3 (11.5%) did not. In eighteen (69.2%) health care facilities, incinerators were well secured by fence. In terms of treatment remnant management, more than two-third of the facilities (68%) maintain the ash in an open area. For ash remnants, only 5 (19.2%) of the facilities use close dumping. Waste disposal sites were discovered far away from any water source in twenty-three (88.4%) of the facilities. In twenty-one (80.7%) of the facilities, separate landfills are used for waste disposal, and sixteen (61.5%) of the facilities had sufficient depth. More than three-quarters of the facilities disinfect liquid waste before disposal, and 24 (92.3%) of the facilities collect it in a septic tank. According to the findings, over two-third of public health facilities (73%) properly implement one or more components of the biomedical waste management system.

**Waste generation in health care facilities.** The mean ±SD of daily solid biomedical waste generation per laboratory in the health facilities was 4.9 ± 3.13 kg/day. Among these, 3.86 ± 2.66 kg/day was an infectious waste, and 1.10 ± 0.853 kg/day was a sharp wastes. Based on the nature of health facilities, hospitals in Addis Ababa were found to generate 3.107 ± 2.016 kg/day of biomedical wastes, in which 5.189 ± 3.807kg/day was infectious wastes and 1.026 ± 0.767 was sharp wastes whereas 20 sampled health centers in the city were found generating 2.331 ± 1.297 kg/day comprising of 3.342 ± 1.762 Kg/day of infectious wastes and 1.102 ± 0.843 kg/day of sharp wastes. A mean of 112 ± 67.5 patients per day gets laboratory service for sampled facilities. Out of this, hospitals were giving laboratory service for 195.8 ± 50.621 patients per day whereas health center had 86.55 ±44.57 patients per day [Table 3].

The research compared the generation rate of biomedical wastes based on the number of patients and found that 2 Hospitals (Zewditu hospital, Triunesh Beijing Hospital) and 5 health centers (Kasanchis health center, Teklehaymanot health center, Mychew health center, Addis ketema woreda 4 health center and Entoto Number 2 health center) had a higher biomedical waste generation rate compared to other sample health institutions [Table 4].

**Factors associated with biomedical waste management.** Age, profession, and both diploma and BSC levels of education were marginally associated with knowledge in multivariate regression analysis. Sex, work experience, and training were discovered to have a statistically significant relationship with knowledge. Males with (AOR: 2.771 95% CI (1.164, 6.596)) and (AOR: 1.559% CI (1.083, 2.244)) have a higher likelihood of knowing common infectious microorganisms associated with poor waste handling and common modes of treatment. Work experience of 1–5 years (AOR: 344 95% CI (19, 6009)), 5–10 years (AOR: 113 95% CI (7.5, 1683.9)) is strongly associated with knowledge of biomedical waste management policies and

**Table 3. Daily laboratory solid waste generation rate in public health care facilities in Addis Ababa, Ethiopia, 2021 (n = 26).**

| Name of health facility | Biomedical waste, kg/day | | | | |
|---|---|---|---|---|---|
| | Patients/ day | Total/kg/day | Mean of Infectious waste | Mean of Sharp waste | Overall mean |
| Goro HC | 200 | 11.5 | 3.5 | 0.3 | 1.91 |
| Ferency Woreda 01 HC | 80 | 3 | 0.76 | 0.23 | 0.49 |
| Entoto kuter 1 HC | 30 | 5.5 | 1.25 | 1 | 1.125 |
| Tirunesh Beijing hospital | 134 | 9.55 | 2.265 | 0.915 | 1.59 |
| Ras desta hospital | 285 | 31 | 10 | 0.33 | 5.17 |
| Mikiland HC | 100 | 12.5 | 3.13 | 0.83 | 1.98 |
| Nifas selk woreda 2 HC | 120 | 8.5 | 2.5 | 0.33 | 1.41 |
| Arada HC | 80 | 7.5 | 1.83 | 0.66 | 1.25 |
| Ghandi memorial hospital | 180 | 6 | 1.66 | 0.33 | 0.995 |
| Menelik II hospital | 206 | 13.65 | 3.91 | 0.63 | 2.27 |
| Bisrate Gebirale HC | 200 | 14.9 | 4.5 | 0.46 | 2.48 |
| Hiwot Amba HC | 60 | 6.2 | 1.7 | 0.33 | 1.015 |
| Zewditu memorial hospital | 200 | 36 | 10 | 2 | 6 |
| Janmeda HC | 63 | 17 | 4.76 | 0.92 | 2.84 |
| Teklehaymanot HC | 70 | 16.7 | 4.43 | 1.13 | 2.78 |
| Kolfe HC | 77 | 20.1 | 5.33 | 1.36 | 3.34 |
| Yekatit 12 hospital | 170 | 10.58 | 3.303 | 1.953 | 2.62 |
| Saris HC | 42 | 13.2 | 3.6 | 0.8 | 4.4 |
| Mychew HC | 80 | 28.4 | 6.13 | 3.33 | 4.73 |
| Kality HC | 75 | 9.5 | 1.66 | 1.5 | 1.58 |
| Addis ketema Woreda 4 HC | 80 | 19.6 | 5.3 | 1.8 | 3.55 |
| Addis ketema woreda 7 HC | 70 | 7 | 1.25 | 1.08 | 1.16 |
| Entoto No.2 HC | 120 | 29 | 6.9 | 2.76 | 4.83 |
| Addis Gebeya HC | 50 | 17.2 | 3.6 | 2.13 | 2.86 |
| Semit HC | 74 | 9.7 | 2.66 | 0.56 | 1.61 |
| Kasanchis HC | 60 | 7.8 | 2.06 | 0.53 | 1.29 |
| Overall mean of Hospitals | 195.8 | 17.7 | 5.189 | 1.026 | 3.107 |
| SD of Hospitals | 50.621 | 12.507 | 3.807 | 0.767 | 2.016 |
| Overall mean of Health centers | 86.55 | 13.24 | 3.342 | 1.102 | 2.331 |
| SD of Health centers | 44.571 | 7.176 | 1.762 | 0.843 | 1.297 |
| Overall mean | 112 | 14.8052 | 3.8653 | 1.1004 | 4.947 |
| SD | 67.5 | 8.75676 | 2.6676 | 0.85389 | 3.1348 |

guidelines in the country. Previous training has been associated to the knowledge of the most common infectious microorganisms and the maximum time of waste [Table 5].

Similarly, the availability of standard operational procedures, the provision of durable waste holding bags, the method of waste transport, the treatment of infectious wastes prior to disposal, the presence of a separate financial source, and legitimately adhering to current biomedical waste management guidelines in medical laboratories are all strongly associated with knowledge of the biomedical waste management system. The concern of laboratory managers about biomedical waste management is also strongly associated with the practice of having standard operational procedures, common modes of treatment, and proper liquid waste management [Table 6].

The number of patients receiving medical laboratory services per day was found to have a statistically significant relationship with the daily total solid waste generation rate per medical laboratory in a linear regression analysis (t = 3.032; 95% CI (2.421, 12.999)) [Table 7].

**Table 4. Comparison of biomedical waste generation rate with number of patients per day getting a laboratory service in public health care facilities (n = 26).**

| Variable | Number of patients | | Name of the organization | Value |
|---|---|---|---|---|
| Solid waste generation Total Kg/day | 60 | Highest | Kasanchis HC | 7.8 |
| | | Lowest | Hiwot Amba HC | 6.2 |
| | 70 | Highest | Teklehymanot HC | 16.70 |
| | | Lowest | Addis Ketema W7 HC | 7 |
| | 80 | Highest | Mychew HC | 28.40 |
| | | | Addis Ketema W4 HC | 19.60 |
| | | Lowest | Ferency W1 HC | 3 |
| | | | Arada HC | 7.5 |
| | 120 | Highest | Entoto No.2 HC | 29 |
| | | Lowest | Nifas selk W02 HC | 8.5 |
| | 200 | Highest | Zewdiut Hospital | 36 |
| | | | Triunesh Beijing Hospital | 19.10 |
| | | Lowest | Goro HC | 11 |
| | | | Bisrate Gebirel HC | 14.90 |

Waste associated with the coronavirus pandemic has a significant positive correlation with waste segregation practice (r = 0.51, p = 0.009) and the availability of color coding (r = 0.431, p = 0.032) [Table 8].

## Discussion

Estimating the rate of biomedical waste generation is critical for health care facilities in order to design and implement a better management system. According to this study, the average mean daily generation rate of biomedical waste in studied medical laboratories in Addis Ababa health care facilities was 4.9 ± 3.13 kg/day per medical laboratory, of which 3.86 ± 2.66 kg/day was hazardous waste and 1.10 ± 0.853 kg/day was sharp waste. According to a study conducted in Adama, Ethiopia, the mean daily generation is 4.46 ± 0.45kg/day per health facility [21]. Because this study only collected data from medical laboratories, the results suggest a higher quantity of waste produced, but other studies now being compared show data from all health facilities [22]. This disparity could be attributed to the fact that medical laboratories

**Table 5. Multivariate logistic regression analysis of factors related to knowledge regarding biomedical waste management.**

| Variable | | Knowledge status | | AOR | 95% Confidence Interval | | P- value |
|---|---|---|---|---|---|---|---|
| | | Yes | No | | Lower Bound | Upper Bound | |
| Gender | Male | 98 | 70 | 2.771 | 1.164 | 6.596 | .021 |
| | Female | 64 | 66 | 0.862 | 1 | | |
| Work experience | <1 year | 14 | 6 | 2.824 | 2.67 | 4.21 | .974 |
| | 1–5 years | 68 | 30 | 344.758 | 19.778 | 512.6 | .001 |
| | 5–10 years | 112 | 22 | 113.050 | 7.590 | 345.6 | .001 |
| | 10–15 Years | 24 | 10 | 605 | 582 | 690 | .991 |
| | 15–20 years | 8 | 2 | 174.8 | 164 | 238 | .983 |
| | >20 years | 2 | 0 | 729.1 | 1 | | |
| Training | Yes | 180 | 2 | 7.290 | 1.574 | 33.767 | .011 |
| | No | 102 | 12 | 0.094 | 1 | | |

**Table 6. Multivariate logistic regression analysis which shows the association of knowledge with the practice of biomedical waste management, Addis Ababa, Ethiopia, 2021.**

| Variable | | Knowledge status | | AOR | 95% Confidence Interval | | P- value |
|---|---|---|---|---|---|---|---|
| | | Yes | No | | Lower Bound | Upper Bound | |
| Having a standard operational procedure | Yes | 204 | 18 | 2.499 | 2.294 | 2.846 | .010 |
| | No | 44 | 14 | 0.862 | | 1 | |
| Waste holding bags enough to resist leak and puncture | Yes | 202 | 18 | 2.587 | 2.341 | 3.010 | .048 |
| | No | 54 | 10 | | | 1 | |
| Method of transport | Use of bare hands | 144 | 30 | 0.998 | 0.743 | 1.341 | .998 |
| | Wheel barrows | 46 | 24 | 1.898 | 1.743 | 2.341 | .045 |
| | Trolley | 42 | 12 | 0.598 | .338 | 1.060 | .078 |
| | Waste container itself | 22 | 2 | | | 1 | |
| Infectious waste treated before disposal | Yes | 206 | 48 | 2.598 | 2.338 | 3.060 | .001 |
| | No | 14 | 20 | | | 1 | |
| Separate financial source | Yes | 64 | 2 | 1.064 | 1.001 | 1.470 | .001 |
| | No | 58 | 22 | | | 1 | |
| | I don't know | 106 | 46 | 0.670 | 0.412 | 1.090 | .107 |
| Following the current policy and guideline legitimately | Yes | 154 | 26 | 1.034 | 1.021 | 1.659 | .001 |
| | No | 48 | 18 | 0.611 | 0.368 | 1.015 | .057 |
| | I don't know | 26 | 26 | | | 1 | |
| **Managers' concern** | | | | | | | |
| Having a standard operational procedure | Yes | 204 | 18 | 2.522 | 2.295 | 2.921 | .025 |
| | No | 28 | 2 | | | 1 | |
| Liquid waste management | Decontaminated before dumping | 216 | 12 | 2.386 | 1.169 | 2.883 | .024 |
| | Dump in to running water | 24 | 8 | 0.648 | 0.429 | 0.978 | .039 |
| | I don't Know | 2 | 10 | | | 1 | |
| Common modes of treatment | Incineration | 132 | 10 | 3.621 | 2.468 | 3.825 | .001 |
| | Autoclaving | 24 | 2 | 1.608 | 0.555 | 4.660 | .382 |
| | Chemical disinfection | 6 | 0 | | | 1 | |

generate a large amount of weight-bearing biological waste, such as glasses, tubes, sharps, and other items.

The rate of biomedical waste generation is determined by the number of patients in the health care facility. According to a study conducted in health care facilities in Addis Ababa, there is a positive linear relationship between the number of patients and the rate of biomedical waste generation [23]. This study also discovered a significant association between the number of patients receiving medical laboratory services per day and the rate of biomedical waste generation, P = 0.006. This indicates that as the laboratory's workload increases, so does the waste generated in that laboratory.

Biomedical wastes generated in health care facilities, including medical laboratories, are hazardous to one's health and the environment and must be managed in accordance with international and national guidelines. According to the findings of this study, there are currently three guidelines regarding health care waste management in Ethiopia, and 63% of laboratory professionals and 81.3% of waste handlers were aware of the guidelines, though 54.1% of them indicated that the guidelines are insufficient for implementing a proper practice. This finding is consistent with the result of a systematic review conducted in Ethiopia, which revealed that regulations in Ethiopia are out of date and that there is a lack of compliance with

**Table 7. Linear regression analysis of the association of number of patients per day with total generation of solid biomedical waste in medical laboratories.**

| Variables | | t- value | 95% Confidence Interval for B | | P- value |
|---|---|---|---|---|---|
| | | | Lower Bound | Upper Bound | |
| Number of patients per day | Solid waste generation Sharp kg/day | -3.516 | -33.006 | -8.474 | .002 |
| | Solid waste generation Total kg/day | 3.032 | 2.421 | 12.999 | .006 |

their implementation [4]. Another study in Ethiopia found that the availability of health care waste management guidelines improves waste handling and management [24]. This study also discovered that knowledge of biomedical waste management policy and guidelines is significantly associated with proper management practice.

Training has been identified as an essential component of effective biomedical waste management. This study discovered that 50.3% of study participants received biomedical waste management training and that it had a significant association with knowledge and proper biomedical waste management practice. This result was better than a finding in Debre Markos, Ethiopia's north-western region, where only 30.9% of study participants were trained in waste management but did not meet national and international standards [14]. Furthermore, this finding is far better compared to a study that found only 2.9% of laboratory professionals were trained in India [25]. Despite the fact that this training rate is inadequate, it does not address the underlying issue and does not lead facilities to proper biomedical waste management and control.

The study found that 96% of study participants confirmed that medical laboratories in health care facilities used color coding-based segregation in a container with a biohazard symbol. The systematic review, on the other hand, mentioned the very limited waste segregation practice [4]. This could be due to an increase in the availability of training and increased awareness among laboratory managers. Furthermore, the coronavirus pandemic had made professionals aware of the importance of segregating pandemic waste, with % of them disposing of it separately in an infectious waste collection bin with a cover. According to the findings of this study, 18.2% of laboratory professionals experienced physical injuries such as needle stick and sharp injury, and 15.6% of them occurred while handling biomedical waste, which is consistent with the findings of a previous study in health facilities of Gonder town [13]. This could be avoided by properly using personal protective equipment, placing safety boxes, and practicing better segregation. Approximately 90.6% of the study participants revealed that they use personal protective equipment for biomedical waste handling. This result is similar to that of the Debre Markos study, which found that 97% of study participants always use PPE when handling biomedical waste [23].

Most medical laboratories in health facilities transport biomedical waste in closed containers, but 39% use open containers, according to another study in Addis Ababa, which found that open containers are used for transporting from collection site to storage and treatment [23, 26]. According to the current study, 56% have a waste storage area and 14.3% are secure. In contrast, 85.7% of the facilities with a storage area are open and unprotected; while 44% have no storage area. A study in Ethiopia discovered a slightly different result, revealing that 40% of health facilities stored biomedical waste in an unprotected environment. The study's

**Table 8. Bivariate correlation analysis of waste of coronavirus pandemic with waste segregation practice and availability of color coding.**

| Variables | | Waste segregation practice | Color coding |
|---|---|---|---|
| Waste of coronavirus pandemic | Pearson Correlation coefficient | .510 | .431 |
| | P-value | .009 | .032 |

findings indicate that health care waste storage practices are deteriorating, posing a significant risk to both health care professionals and the environment [27].

## Limitation of the study

Despite revealing the generation rate and management practices of biomedical wastes from clinical laboratories in Ethiopia that generate dangerous infectious wastes and data is scarce, the study was limited to measuring solid biomedical wastes. The study was unable to include the generation rate and management of liquid biomedical waste management due to financial and other limitations, such as a lack of measuring materials. As a result, we recommend additional research on the generation rate and management system of liquid biomedical waste in health facilities in Addis Ababa, Ethiopia, to gain a better understanding of the burden and identify gaps that can be used to design intervention mechanisms.

## Conclusion

The study revealed that medical laboratories generate a higher amount of biomedical waste per day, which is significantly related to the number of patients who receive laboratory services. There is a lack of understanding of biomedical waste management policies and guidelines, as well as common types, components, and infectious microorganisms associated with biomedical wastes and their management. The training was found to have a significant association with proper knowledge and practice of biomedical waste management. This study found proper waste segregation, better waste storage, and treatment practices for hazardous and non-hazardous waste. In contrast, insufficient and inappropriate transportation equipment was observed, and open burning pits were commonly used for final waste disposal.

For improved practice, Ethiopia's current national policy and guidelines for biomedical waste management should be revised and strengthened. Furthermore, increased attention from laboratory and facility managers is critical in order to efficiently implement the standard and provide frequent training. Further research into biomedical waste management practice, such as comparing each health care facility's SOP with national and international guidelines, liquid biomedical waste management, and COVID-19 related waste management, is also recommended.

## Supporting information

**S1 File. Questioner used for to assess the generation rate and management system of biomedical wastes and their associated factors in medical laboratories.**
(DOCX)

## Acknowledgments

We would like to acknowledge Addis Ababa University for giving us the opportunity to conduct the study. Our gratitude also goes to all data collectors and health facility mangers for their unreserved cooperation. Last but not least, we are indebted to the study participants without whom this study would not be realized.

## Author Contributions

**Conceptualization:** Salem Endris, Abay Sisay.

**Data curation:** Salem Endris, Zemenu Tamir, Abay Sisay.

**Formal analysis:** Salem Endris, Zemenu Tamir, Abay Sisay.

**Funding acquisition:** Salem Endris, Zemenu Tamir, Abay Sisay.

**Investigation:** Salem Endris, Zemenu Tamir, Abay Sisay.

**Methodology:** Salem Endris, Zemenu Tamir, Abay Sisay.

**Project administration:** Salem Endris, Zemenu Tamir, Abay Sisay.

**Resources:** Salem Endris, Zemenu Tamir, Abay Sisay.

**Software:** Salem Endris, Zemenu Tamir, Abay Sisay.

**Supervision:** Zemenu Tamir, Abay Sisay.

**Validation:** Salem Endris, Zemenu Tamir, Abay Sisay.

**Visualization:** Zemenu Tamir, Abay Sisay.

**Writing – original draft:** Salem Endris, Zemenu Tamir, Abay Sisay.

**Writing – review & editing:** Salem Endris, Zemenu Tamir, Abay Sisay.

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
