## [Decision Letter · Decision Letter 0]

11 Jan 2022

PONE-D-21-25475Medical laboratory waste generation rate, management practices and associated factors in Addis Ababa, EthiopiaPLOS ONE

Dear Dr. Sisay,

Thank you for submitting your manuscript to PLOS ONE. After careful consideration, we feel that it has merit but does not fully meet PLOS ONE’s publication criteria as it currently stands. Therefore, we invite you to submit a revised version of the manuscript that addresses the points raised during the review process.Please see below the comments and suggested MAJOR revisions made by the individual(s) who reviewed your manuscript.  If provided, the referee's report(s) indicate the revisions that need to be made before it can be accepted for publication.

We look forward to receiving your revised manuscript.

Kind regards,

Ricardo Santos

Academic Editor

PLOS ONE

Journal Requirements:

Whilst you may use any professional scientific editing service of your choice, PLOS has partnered with both American Journal Experts (AJE) and Editage to provide discounted services to PLOS authors. Both organizations have experience helping authors meet PLOS guidelines and can provide language editing, translation, manuscript formatting, and figure formatting to ensure your manuscript meets our submission guidelines. To take advantage of our partnership with AJE, visit the AJE website (http://aje.com/go/plos) for a 15% discount off AJE services. To take advantage of our partnership with Editage, visit the Editage website (www.editage.com) and enter referral code PLOSEDIT for a 15% discount off Editage services.  If the PLOS editorial team finds any language issues in text that either AJE or Editage has edited, the service provider will re-edit the text for free.

4. Please improve statistical reporting and refer to p-values as "p<.001" instead of "p=.000". Our statistical reporting guidelines are available at https://journals.plos.org/plosone/s/submission-guidelines#loc-statistical-reporting.

We would like to acknowledge Addis Ababa University for giving us the opportunity and financing the research

This research was financed by Addis Ababa University. However, the funder had not any involvement with the research methodology design, analysis and write up of the manuscript.

Reviewers' comments:

Reviewer's Responses to Questions

**Comments to the Author**

1. Is the manuscript technically sound, and do the data support the conclusions?

Reviewer #1: Partly

Reviewer #2: Yes

Reviewer #3: Partly

2. Has the statistical analysis been performed appropriately and rigorously? 

Reviewer #1: Yes

Reviewer #2: Yes

Reviewer #3: I Don't Know

3. Have the authors made all data underlying the findings in their manuscript fully available?

Reviewer #1: Yes

Reviewer #2: Yes

Reviewer #3: No

4. Is the manuscript presented in an intelligible fashion and written in standard English?

Reviewer #1: No

Reviewer #2: Yes

Reviewer #3: No

5. Review Comments to the Author

Reviewer #1: Overall, this study needs further work (in writing), maybe could benefit from editing by a professional editor whose primary speaking and writing is English.

I thought some basic information are missing in the introduction and discussion. The author did not provide information (that I saw-maybe I missed it?) on the type of medical waste, and international standards on medical waste management such as color coding and symbolled containers/bags. Also needs further detailed discussion.

It is unclear how potential participants were contacted by the primary investigator, how many participants were identified (inclusion criteria need to be mentioned) and how many were offered to participate in the study. The author did not clarify at what point data saturation was reached?

Regarding conceptual/theoretical framework, the study design needs further description of its theoretical framework and principles.

Sending this study back to author with these suggestions could be promising.

Reviewer #2: I would like to thank the authors for tackling a very important issue with the example of Addis Ababa.

It was exciting to read the work.

As a suggestion, I think that the discussion section should be examined in comparison with the results of other countries.

Reviewer #3: The authors have submitted a manuscript that details a research project designed to evaluate the generation rate of biomedical waste, management practices and associated factors among public healthcare medical laboratories in Addis Ababa, Ethiopia. The topic is an important one across the globe and especially important in developing countries where resources are scarce. Public health officials, members of the health care delivery systems, administrators and the public have an interest in this evaluation. Unfortunately, the presentation does not do justice to the amount of work performed and the underlying value of the data generated. Although a considerable amount of useful data has been generated, the manuscript could be improved to provide more explicit information. Finally, the discussion section could be improved to provide the reader with the important take away messages generated by the research.

6. PLOS authors have the option to publish the peer review history of their article (what does this mean?). If published, this will include your full peer review and any attached files.

Reviewer #1: **Yes: **Muhammad Al-Haddad

Reviewer #2: No

Reviewer #3: No

---

## [Author Response · Author response to Decision Letter 0]

9 Feb 2022

Dear Editor: 

We are very much glad to write this response that our research output manuscript entitled “Medical laboratory waste generation rate, management practices and associated factors in Addis Ababa, Ethiopia, with a manuscript reference number PONE-D-21-25475” has been possible considered for publication in PLOS ONE. We are pleased to have an opportunity to make our manuscript revised and we have greatly appreciated for the reviewers’ high level comments, and suggestions were very helpful for the overall improvement of the manuscript. In revising the manuscript, we have carefully considered reviewers’ comments, academic editor and suggestions on our revised submission. 

As instructed, we have attempted to concisely explain changes made in reaction to all comments and reply to each comment in point-by-point fashion as appended here

The detail is attached in the system as rebuttal letter

---

## [Decision Letter · Decision Letter 1]

30 Mar 2022

Medical laboratory waste generation rate, management practices and associated factors in Addis Ababa, Ethiopia

PONE-D-21-25475R1

Dear Dr. Sisay,

We’re pleased to inform you that your manuscript has been judged scientifically suitable for publication and will be formally accepted for publication once it meets all outstanding technical requirements.

Kind regards,

Ricardo Santos

Academic Editor

PLOS ONE

Additional Editor Comments (optional):

Reviewers' comments:

Reviewer's Responses to Questions

**Comments to the Author**

1. If the authors have adequately addressed your comments raised in a previous round of review and you feel that this manuscript is now acceptable for publication, you may indicate that here to bypass the “Comments to the Author” section, enter your conflict of interest statement in the “Confidential to Editor” section, and submit your "Accept" recommendation.

Reviewer #1: All comments have been addressed

Reviewer #2: (No Response)

2. Is the manuscript technically sound, and do the data support the conclusions?

Reviewer #1: Yes

Reviewer #2: Yes

3. Has the statistical analysis been performed appropriately and rigorously? 

Reviewer #1: Yes

Reviewer #2: Yes

4. Have the authors made all data underlying the findings in their manuscript fully available?

Reviewer #1: Yes

Reviewer #2: Yes

5. Is the manuscript presented in an intelligible fashion and written in standard English?

Reviewer #1: Yes

Reviewer #2: Yes

6. Review Comments to the Author

Reviewer #1: Comments were adequately addressed and the manuscript complies with the journal and publication criteria. Also, the flow of information and details are now presented in a logical and smooth way. No additional comments.

Reviewer #2: good study. I think it will contribute to the literature. congratulations to the authors.

acceptable.

7. PLOS authors have the option to publish the peer review history of their article (what does this mean?). If published, this will include your full peer review and any attached files.

Reviewer #1: **Yes: **Muhammad S. Al-Haddad

Reviewer #2: No

---

## [Editor Report · Acceptance letter]

7 Apr 2022

PONE-D-21-25475R1 

Medical laboratory waste generation rate, management practices and associated factors in Addis Ababa, Ethiopia 

Dear Dr. Sisay:

I'm pleased to inform you that your manuscript has been deemed suitable for publication in PLOS ONE. Congratulations! Your manuscript is now with our production department. 

Kind regards, 

on behalf of

Dr. Ricardo Santos 

Academic Editor

PLOS ONE